# Detection of virus-neutralising antibodies and associated factors against rabies in the vaccinated household dogs of Kathmandu Valley, Nepal

Shikha Rimal[1,2], Krishna Chandra Ojha[1], Warangkhana Chaisowwong[2,3], Yogendra Shah [1], Dhan Kumar Pant[1,4,5], Anucha Sirimalaisuwan[2,3]*

1 National Zoonoses and Food Hygiene Research Centre, Kalimati, Nepal, 2 Veterinary Public Health and Food Safety Centre for Asia Pacific and Excellent Center of Veterinary Public Health, Faculty of Veterinary Medicine, Chiang Mai University, Chiang Mai, Thailand, 3 Department of Veterinary Biosciences and Veterinary Public Health, Faculty of Veterinary Medicine, Chiang Mai University, Chiang Mai, Thailand, 4 Institute of Medicine, Teaching Hospital, Tribhuwan University, Maharajgunj, Kathmandu, Nepal, 5 National Academy of Medical Sciences, Kathmandu, Nepal

* anucha.siri@gmail.com

**Data Availability Statement:** All relevant data are within the paper and its additional supporting information files.

## Abstract

### Background

Rabies is a vaccine-preventable neglected tropical viral zoonosis. It occurs worldwide, creating a very heavy burden in many developing countries, including Nepal. Dogs are the principle vector for the transmission of this disease in urban areas. Vaccination is the most important preventive measure in areas where dogs are the principle source of infection. This study was conducted with the aim of detecting virus-neutralising antibodies and associated factors against rabies in vaccinated household dogs of Kathmandu valley.

### Methods

Blood samples were collected from 110 vaccinated pet dogs in Kathmandu, Bhaktapur, and Lalitpur districts of Nepal. The samples were taken to the laboratory of the National Zoonosis and Food Hygiene Research Center where serum was separated. An indirect immune-enzymatic assay (*Platelia*$^{TM}$ *Rabies II kit ad usum Veterinarium*, Biorad, China) was used for the detection of rabies virus anti-glycoprotein antibodies in the dog serum samples following the manufacturer's recommendations and instructions. Optical density values for unknown samples were compared with the positive sera titers in quantification tests obtained after a direct reading on the standard curve. Results were expressed as equivalent units per ml (EU/ml).

### Findings

Of the total samples, 89.09% exceeded the required seroconversion level ($\geq$ 0.5 EU/ml); another 9.09% did not reach the seroconversion level (0.125–0.5 EU/ml); and 1.81% had undetectable seroconversion levels (<0.125 EU/ml) suggesting that the animal had not

**Funding:** This study was supported by the Veterinary Public Health and Food Safety Centre for Asia Pacific and Excellent Center of Veterinary Public Health, Faculty of Veterinary Medicine, Chiang Mai University, Chiang Mai, Thailand to Anucha Sirimalaisuwan (A.S) with Grant award no: R000016652.

**Competing interests:** The authors have declared that no competing interests exist.

seroconverted according to the PLATELIA™ RABIES II test. Only one factor, the condition under which the dog was kept, was significantly associated with the antibody titer level. No association was found for any of the other factors included in the study.

## Interpretation

Vaccination is the most effective measure for prevention and control of rabies. The locally manufactured brand of vaccine, which is available in Nepal, is potent enough to generate a sufficient amount of protective antibodies, equal to international brands.

## Introduction

Rabies is a vaccine-preventable, viral zoonosis belonging to neglected tropical diseases that can infect all mammals, but over 99% of all human death from rabies are caused by domestic dogs [1]. According to the World Health Organization (WHO), 59,000 people die from this disease yearly, of which 30–50% are below 15 years of age. Over 95% of the human rabies cases are concentrated in Asia and Africa, with Asia having 60% of the total number of cases [2] [3]. Dog bites are the most important cause of human deaths from rabies in those regions [4] [5] [6].

Nepal is a landlocked country between China and India; the latter having the largest rabies burden in the world [3] [7]. Nepal has an open border with India and the socioeconomic status is quite similar in the two countries. China has the second highest incidence of rabies [8] with an annual 2,000 to 3,000 human deaths, of which more than 95% are due to the bite of a rabid dog [5]. With the two neighboring countries having a high rabies burden, the disease has become endemic and a priority zoonotic disease in Nepal as well [9]. There, the disease has been confirmed in cattle, buffaloes, goats, alpacas, dogs, and mongooses. Rabies occurs throughout the year, and dogs are the principle vector for the transmission of disease. Unfortunately, the true status of rabies burden due to vectors in Nepal is unknown.

There are two primary and interrelated epidemiological cycles of rabies in Nepal. One is the urban cycle which involves domesticated dogs, and the other is the sylvatic cycle which involves wildlife [10]. A study conducted between 1991 and 2000 showed that more than 96% of rabies patients shared a history of rabid dog exposure, indicating that the urban cycle is the main source of human rabies [9]. Large numbers of human rabies cases in Nepal are due to stray and community dogs. Host wise prevalence study also suggests that 61.7% rabies cases were detected in stray and community dogs [11]. Thus, this rabies cycle is maintained by stray and community dogs. Sometimes there is spill-over from stray and community dogs to pet dogs, which further adds to the human rabies burden.

Vaccination of dogs is one of the most important preventive measures in areas where dogs are the main source of human infection. For the dog rabies mass vaccination program, achieving a coverage of least 70% appears to be sufficient to prevent transmission rabies to humans for at least 6 years [10,11]. A vaccination rate of 60–70% of pet dogs has been shown to result in a drastic decrease in the prevalence of the disease in humans [12]. As vaccination is the primary control measure, it is important to determine the level of anti-rabies antibodies in animals to evaluate the efficacy of the control measures [13].

So far, no studies have been conducted in Nepal to evaluate the efficacy of the vaccines being used, or to determine if the vaccination programs that have been implemented are an effective strategy for the prevention and control of rabies. Various studies have shown that the

type of vaccine used, the number of vaccinations, the interval between vaccinations and blood sample collection, age at vaccination, size and breed of dog can influence the antibody response, [14] but that has not been investigated in Nepal. The present study aimed to detect the antibody titer level and associated factors in vaccinated pet dogs to determine the effectiveness of the Nepali vaccination program as a method of prevention of rabies.

## Materials and methods

### Ethical consideration

Ethical approvals were taken from the Nepal Veterinary Council and Nepal Health Research Council in context of animal and human (Ref No: 293-2073/74 and Ref No: 759) (S6 and S7 Files). No human intervention was conducted in this study. Written consent was taken from the dog owners before conducting the research (S4 File). For the blood sample collection, agriculture animal use protocol of Chiang Mai University, Thailand was followed (Ref No: FA001/2560[02/2560-08-10]) (S8 File). No anesthesia or analgesia was given to the pet dogs.

### Study design and sample calculation

A cross-sectional study was conducted involving 110 vaccinated pet dogs in Kathmandu, Bhaktapur and Lalitpur districts of Kathmandu valley, shown in Map 1. Inclusion criteria were a dog ages above 4 months, receipt of at least one anti-rabies vaccination, no fever or external injury at the time of blood collection, and blood collection done at least one month after the last booster.

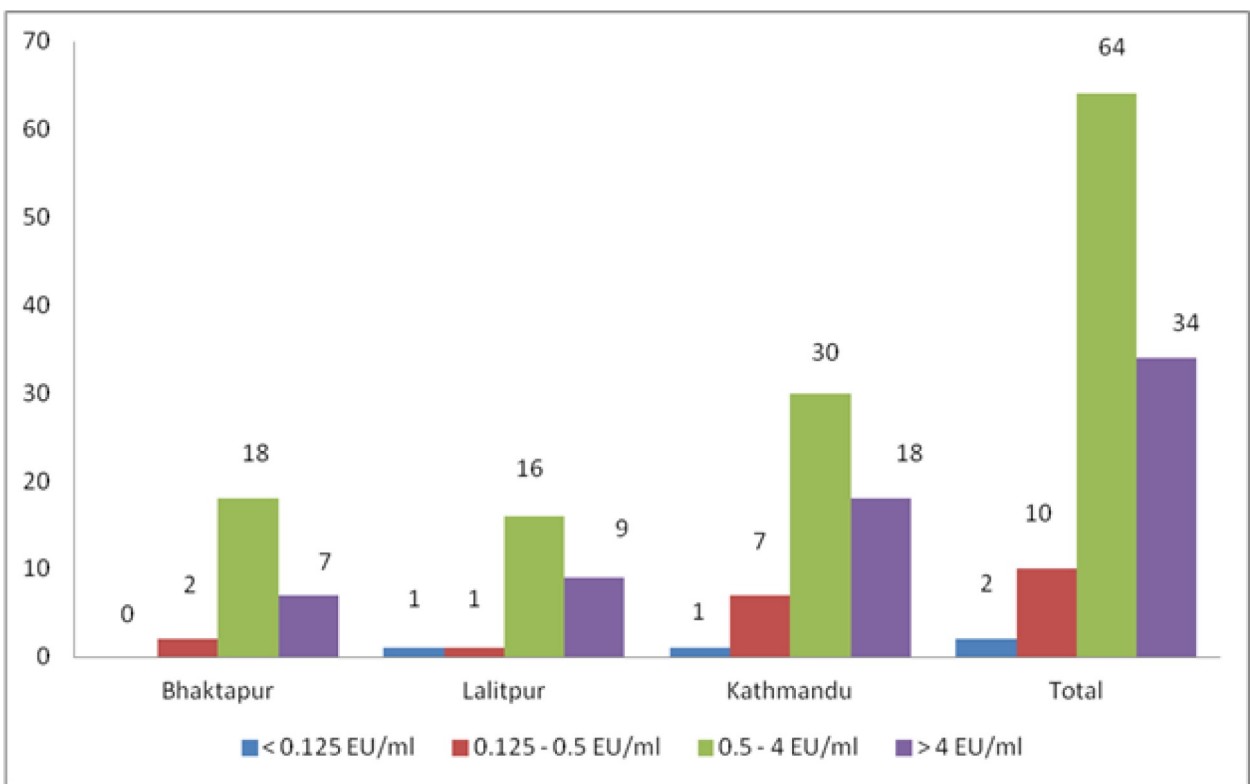

**Map 1. Sampling sites and sampling locations in three selected districts of Kathmandu Valley.**

Blood samples were collected from either the cephalic veins or saphenous vein for antibody detection. Blood was drawn aseptically following the animal use protocol. Samples were kept in an ice box and transferred to the laboratory of the National Zoonosis and Food Hygiene Research Center. The blood samples were allowed to clot and then centrifuged at 2000 rpm for 15 minutes. The serum sample was separated and stored in a cryovial at -18 to -20° centigrade until further testing.

## Antibody titre detection using ELISA–Platelia$^{TM}$ Rabies II kit ad usum Veterinarium

For detection of rabies virus anti-glycoprotein antibodies in the dog serum samples, an indirect immune-enzymatic assay (*Platelia$^{TM}$ Rabies II kit ad usum Veterinarium*, Biorad, China) was used, following the manufacturer's recommendations and instructions. This test was chosen for rapidity in obtaining results (<3 hours), simplicity in comparison to virus neutralization, the quantitative and qualitative results, and safety for the laboratory personnel. This test had 98.6% specificity and 88.8% sensitivity.

The serum samples were diluted into 1/10 ratio (10 μl of sample in 990 μl of dilution solution). The diluted serum samples, the positive and negative controls, and the quantification standard were distributed into microplates and then incubated at 37°C for one hour. To remove unbound antibodies and other proteins in the samples after incubation, three washing steps were performed. Then 100 μl conjugate-protein A labeled with peroxidase was added to each well, followed by a second incubation at 37°C for one hour and an additional five washing steps to remove unbound conjugate. The presence of the immune complexes was highlighted by adding to each well, a peroxidase substrate solution and a chromogen followed by incubation at room temperature for 30 minutes and the addition of 100μl solution of $H_2SO_4$ 1N to stop the enzymatic reaction. The microplates were read bichromatically at 450 and 620 nm.

For the quantitative determination of anti-rabies antibodies, a standard curve was constructed using the quantification standards (S1 to S6 Files), obtained by serial dilutions of the R4b calibrated positive controls. The optical density values for the unknown samples were compared with the positive sera titers in quantification tests, obtained after a direct reading on the standard curve and expressed as equivalent units per ml (EU/ml), a unit equivalent to the international units defined by seroneutralization. The results were categorized as high seroconversion level (>4 EU/ ml), sufficient seroconversion level (0.5–4 EU/ml), insufficient seroconversion level (0.125–0.5 EU/ ml), and undetectable seroconversion (<0.125 EU/ml).

### Data collection

A questionnaire was used to gather information regarding each pet dog (age, sex, breed), vaccination details (boosters given or not, age at booster, health status during vaccination, place of vaccination, person who conducted the vaccination, how many vaccines were given together), and dog management (whether the dog lives in the owner's house or not, whether the dog is restrained or allowed to roam, food given, if they trained or untrained, and, if trained, by whom). (S3 File)

### Data analysis

Data analysis was done using the R Foundation for Statistical Computing Software (R version 3.3.2 (2016-10-31)). For the rabies antibody titer, descriptive statistics was applied and proportions, standard curve, and $R^2$ were derived. For the factors potentially associated with rabies, analytical statistics (chi-square test, odds ratio) were applied and P-values were calculated. Factors with P-values <0.05 were listed as the associated factors.

**Table 1. Results of dog serum samples by district in Kathmandu Valley.**

| District | No. of Samples | Positive/ Negative | Criteria Result Validation |
|---|---|---|---|
| **Bhaktapur** | 2 | - | Not Seroconverted |
| | 25 (92.59%) | + | Seroconverted |
| **Kathmandu** | 8 | - | Not Seroconverted |
| | 48 (85.71%) | + | Seroconverted |
| **Lalitpur** | 2 | - | Not Seroconverted |
| | 25 (92.59%) | + | Seroconverted |
| **Total** | **110** | **- 12 (10.91%)** | - |
| | | **+ 98 (89.09%)** | |

- : Negative

+ : Positive

## Results

### Qualitative results

The obtained serum antibody titer levels were compared with the WHO recommended level of protection ($\geq$ 0.5 IU/ml). The district-wise prevalence of positive results for dog serum is shown in Table 1.

### Quantitative results

To determine the quantity of anti-rabies antibodies in each sample, the optical density compared to a standard curve. The serum titer of all samples was obtained after a direct reading on the standard curve and was expressed as Equivalent Units per milliliter (EU/ml), representing the quantitative determination.

Out of 110 samples from Kathmandu valley, 89.09% samples met or exceeded the required antibody titers level ($\geq$ 0.5 EU/ml), another 9.09% did not reach the antibody titers level (0.125–0.5 EU/ml), and 1.81% samples had undetectable antibody titers ($<$0.125 EU/ml) according to PLATELIA™ RABIES II test (Fig 1). The $r^2$ for both Plate 1 and 2 was 0.97 respectively. (S1 and S2 Files)

Many types of commercial rabies vaccine are available in the Nepalese market. Three of the most popular vaccines such as Defensor (Killed virus strain), Biocan R (Inactivated virus strain) and NeJa Rab (Inactivated cell culture rabies vaccine) were included in the research. The titer levels of the three different types of vaccine within a period of one year of vaccination are shown in Fig 2. The geometric mean and SD values for vaccine A, B and C were 1.96 and 1.30; 2.36 and 1.43; 2.74 and 1.26 respectively. However, there was no significant difference between these three vaccines.

The overall failure rate of the different vaccines varied from 3 to 17%. The vaccine B had high failure rate (17%). High vaccine failures were observed in dogs less than two years old (47%) followed by dogs over 7 years (33%). The highest rate of vaccine failures was observed in crossbreeds (55%), followed by exotic breeds (28%), and local breeds (17%). Medium size breeds (12.9%) showed higher failure rates compared to smaller breeds (9.09%) and larger breeds (8.1%).

Month wise comparison of antibody titer in dogs vaccinated with two different vaccines, A and C, is shown in Fig 3. Vaccine B was excluded from this comparison due to lack of data for all the months. The results confirmed the immunogenicity of both the vaccines. Both vaccines induced high seroconversion within a month of the booster vaccination.

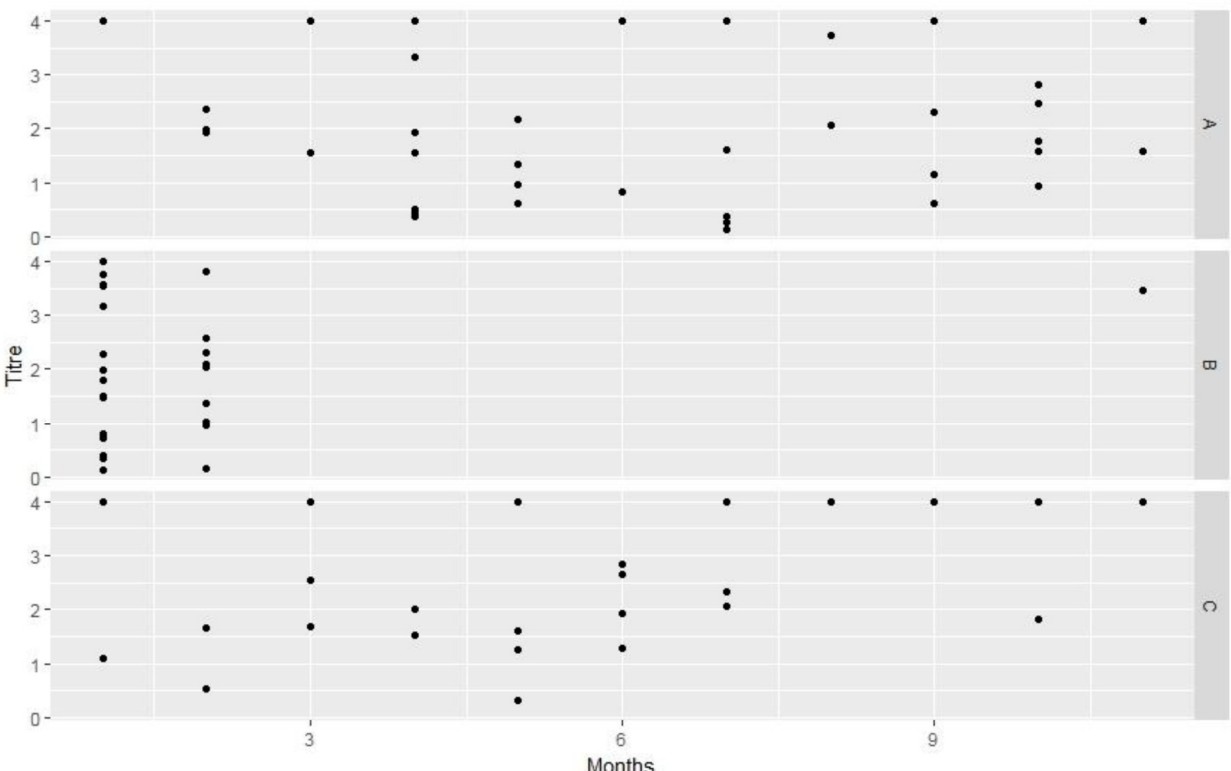

**Fig 1. Quantity of anti-rabies antibodies in 110 dog serum samples determined by comparing the optical density of the sample to a standard curve (>4 EU/ml = high seroconversion level; 0.5–4 EU/ml = sufficient seroconversion level; 0.125–0.5 EU/ml = insufficient seroconversion level; <0.125EU/ml = undetectable seroconversion).**

### Factors associated with rabies antibody titer level

Twenty eight different factors were analyzed univariately to identify associations between the factors and antibody titer level. Factors having P- values < 0.05 are shown in Table 2. Only one factor (how the dog is kept) was signficantly association with rabies antibody titer level.

## Discussion

In this study, virus neutralizing antibody was detected in 110 pet dogs which had been vaccinated against rabies. It was found that out of the 110 vaccinated dogs, 89.09% had the necessary protective level of anti-rabies antibodies ($\geq$0.5 IU/mL), which is in line with a study conducted by Berndtsson et al. [14] that reported 91.9% had a certified test result of $\geq$ 0.5 IU/ mL. The current study found a higher percentage of dogs had a protective level of anti-rabies antibodies than that reported in a study by Singh et al. [15] which showed only 16% pet dogs in Chandigarh, India, had a protective level of anti-rabies antibodies. Millan et al. [16] found a seroprevalence of 20% in Uganda, while Kitala et al. [17] reported 21% in Kenya. In Nigeria, Mauti et al. [4] found that 43% of dogs had antibody titers exceeding the protective threshold.

The influencing factors associated with antibody titers level and their P-value are presented in Table 2. Of the 110 dogs, 48 dogs had been vaccinated with vaccine A, 31 with vaccine B, and 31 with vaccine C. All three vaccines A, B and C induced a high seroconversion within a month of the booster vaccination. This is similar to the findings of Minke et al. [18] who reported a peak antibody response in pets between 3 to 6 weeks after vaccination. Poor

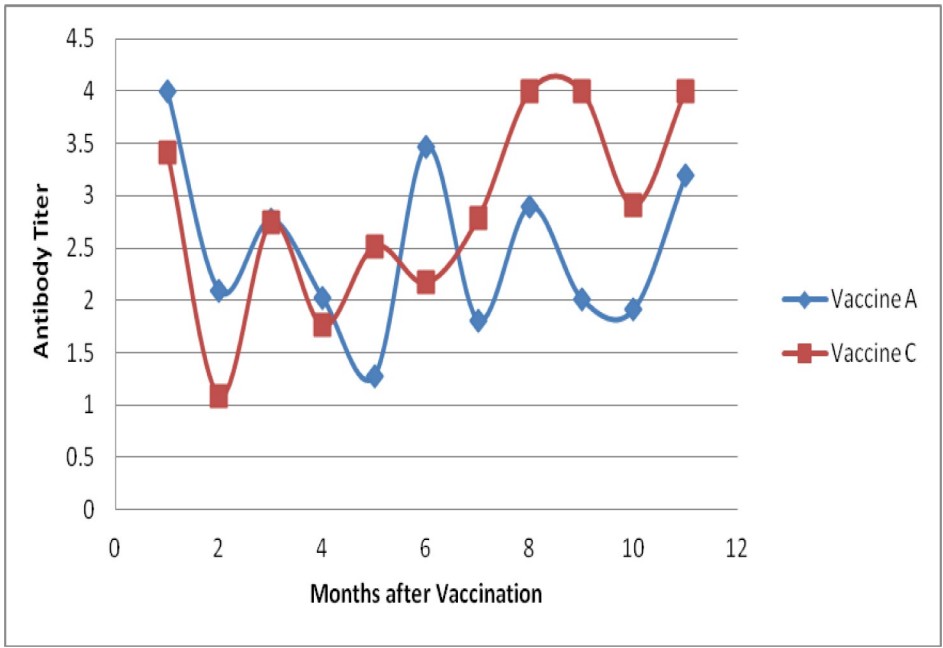

**Fig 2. Three commercially available vaccines and the antibody titer in dogs by months after vaccination.** * Titre: Rabies antibody Titre. Months: The months after which the samples were collected. A, B, C: Different types of Vaccines used.

response of vaccine B (17% failure rate) may be due to the age of the vaccinated pet dogs (as 3/31 dogs were below the age of two years and 2/31 dogs were above seven years). Also the formulation and concentration of vaccine B was different in comparison to the other two

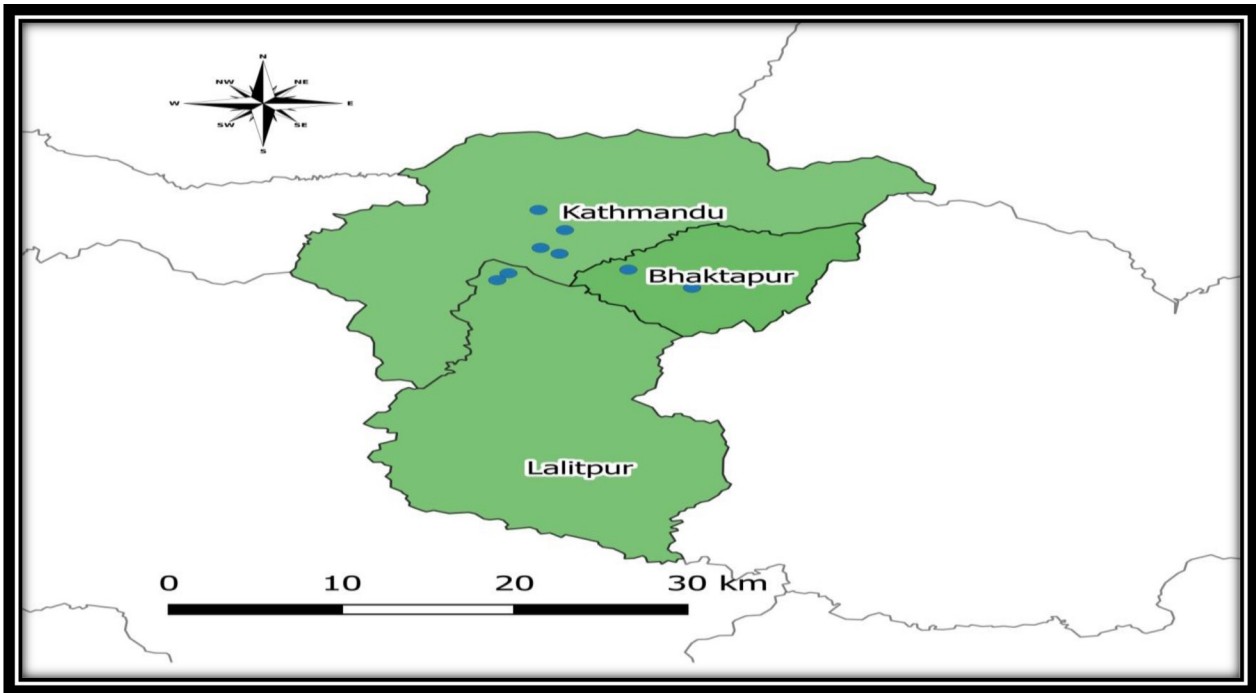

**Fig 3. Month by month comparison of median antibody titers with vaccines A and C.**

**Table 2. Factors associated with rabies antibody titer level.**

| Factors | Sub factors | T- (%) | T+ (%) | OR | 95% CI | P- Value |
|---------|-------------|--------|--------|-----|--------|----------|
| **Age** | > 2 yrs | 8 (7.27%) | 89 (80.9%) | 0.2 | 0.04–1.12 | 0.01 |
| | 2 yrs or less | 4 (3.63%) | 9 (8.18%) | | | |
| **Place where dog is kept** | Indoor | 0 | 26 (23.63%) | 0 | 0–1.08 | 0.04 |
| | Outdoor | 12 (10.9%) | 72 (65.45%) | | | |
| **How dog is kept** | Free | 6 (5.45%) | 81 (73.63%) | 0.21 | 0.05–0.90* | 0.008 |
| | Not Free | 6 (5.45%) | 17 (15.45%) | | | |
| **Provided Training** | No | 8 (7.27%) | 87 (79.09%) | 0.25 | 0.05–1.36 | 0.03 |
| | Yes | 4 (3.63%) | 11 (10%) | | | |

* Has association significant

T- : Negative Treatment

T+ : Positive Treatment

OR : Odds Ratio

CI: Confidence Interval

vaccines. However, there was no significant relationship between the type of vaccine used and the antibody titer level obtained. One of the vaccines included in the study was commercially produced in Nepal. All three commercially available vaccines had the same potency to generate a sufficient protective level of anti-rabies antibody ($\geq$ 0.5 IU/mL), indicating that the Nepalese anti-rabies vaccine was not significantly different from the imported vaccines in terms of generating immunogenicity in dogs.

Higher numbers of vaccine failures in the dogs below two years of age may be due to poor immune response in younger dogs, because their immune system is less well developed compared to older dogs. A similar decline in immune response in older humans has been documented [19], which may explain the higher failure rates in older dogs. Life expectancy is another complicating factor, i.e., "old" and "young" varies according to the breed as well. In the univariate analysis, however, no significant relationship between the age of the dog and the titer level was found.

Vaccination failure rates were higher in cross-breeds than in exotic and local breeds. Univariate analysis did not find any significant differences between breed and titer, similar to the findings of Jakel et al. [20] and Berndtsson et al. [14]. Higher failure rates were observed in medium size dogs compared to small and large dogs. Kennedy et al. [21] reported that most failures occurred in larger breeds, while some smaller breeds also showed higher failure rates. But in this study, there was no relationship between the size of the dog and the antibody titer level. The higher failure rates in cross-breeds may be due to the genetic diversity resulting from cross breeding in different countries.

In Kathmandu valley, there is a general practice to give core vaccine to pet dogs which includes anti- rabies vaccine along with canine distemper combined vaccine DHPPiL (which includes vaccines for canine distemper, hepatitis, parvo virus, para influenza and leptospirosis), while corona virus vaccination is optional. However, there was no difference in antibody titer level or vaccine performance with two or more vaccines given at the same time (anti-rabies and corona, anti-rabies and canine distemper combined vaccine DHPPiL or anti-rabies, corona and canine distemper combined vaccine DHPPiL together). As yet, no studies on variation in canine parvovirus, has been published from Nepal. Mansfield et al. [22] and Jakel et al. [20] found differences in antibody response related to gender, but this study found no such differences, which was similar to the findings of Berndtsson et al. [14].

Vaccine failure was observed in twelve dogs in this study. Ten of those dogs had an insufficient titer level, while two dogs had an undetectable titer level. Those results could be due to influencing factors like age, breeds, type of vaccines, poor administration, out of date and poor storage etc. Vaccines produced by different manufacturers have significantly different failure rates, and significantly different median titers of response. This is presumably due to their formulation and the production differences between vaccines together with the concentration and integrity of antigen content and the adjuvant used [21].

The twelve dogs with low antibody titer were revaccinated and titer tested again. It was found that most people preferred to keep their dogs tied up outside the house, allowed to roam free, or caged outside the house. Dogs that were not kept at home had a greater chance to interact with stray dogs and wildlife that could be risk of infection for other animals. Dogs which stayed outdoors were comparatively less well cared for than the dogs which stayed inside and had frequent contact with their owners. Due to some health conditions, prior to the vaccination or after the vaccination, the immune system in some dogs might have been compromised which led to a delayed immune response. Among the five classes of immunoglobulins, IgG has the predominant role in protecting against infections. Some animals have normal levels of immunoglobulins but does not produce sufficient, specific IgG antibodies, while others lack the ability to produce protective IgG antibodies against specific diseases due to abnormalities in their genetic makeup [23].

## Implications of the research

Based on the results from the research, all the vaccines showed similar potency and efficacy. However, in the market, there is a tendency to charge extra for some vaccines, claiming that the vaccine has higher potency in comparison to the other commonly available vaccines. It can be said that the consumers are free to choose any of the three available vaccines without any risk. The antibody titer level after one year of vaccination is sufficient to fight against rabies. Therefore, it can be safely said that anti-rabies vaccination should be done yearly.

## Conclusions

Although only one factor examined was found to have a significant association with the rabies antibody titer, it is important to realize that vaccination is one of the most important measures for the prevention and control of rabies. By breaking the disease transmission cycle, rabies can be controlled in the dog population and simultaneously, reduce the human rabies burden as well. Based on the results of this study, yearly booster vaccination of dogs is recommended. Titer testing post vaccination would aid in evaluating the efficacy of the vaccines. Further study is necessary to be conducted in the stray and community dog population.

## Supporting information

**S1 Data.**
(XLSX)

**S1 File. Optical densities values for ELISA plate 1 and 2.**
(DOCX)

**S2 File. Antibody titer for ELISA plate 1 and 2.**
(DOCX)

**S3 File. Questionnaire used for data collection.**
(DOCX)

**S4 File. Consent form for blood collection.**
(DOCX)

**S5 File. Consent form translated to local language (Nepali).**
(DOCX)

**S6 File. Animal ethical clearance letter.**
(DOCX)

**S7 File. Human ethical clearance letter.**
(DOCX)

**S8 File. Certificate of approval for use of laboratory animals at laboratory animal center.**
(DOCX)

## Acknowledgments

The authors would like to acknowledge the staffs of National Zoonoses and Food Hygiene Research Center for their kind support during the research period.

## Author Contributions

**Conceptualization:** Shikha Rimal, Warangkhana Chaisowwong, Anucha Sirimalaisuwan.

**Data curation:** Shikha Rimal, Krishna Chandra Ojha, Warangkhana Chaisowwong, Yogendra Shah, Dhan Kumar Pant, Anucha Sirimalaisuwan.

**Investigation:** Shikha Rimal, Krishna Chandra Ojha, Warangkhana Chaisowwong, Dhan Kumar Pant, Anucha Sirimalaisuwan.

**Methodology:** Shikha Rimal, Warangkhana Chaisowwong, Dhan Kumar Pant, Anucha Sirimalaisuwan.

**Project administration:** Shikha Rimal, Krishna Chandra Ojha, Warangkhana Chaisowwong, Yogendra Shah, Dhan Kumar Pant, Anucha Sirimalaisuwan.

**Software:** Shikha Rimal.

**Supervision:** Dhan Kumar Pant.

**Visualization:** Shikha Rimal, Warangkhana Chaisowwong, Dhan Kumar Pant, Anucha Sirimalaisuwan.

**Writing – original draft:** Shikha Rimal, Warangkhana Chaisowwong, Yogendra Shah, Dhan Kumar Pant, Anucha Sirimalaisuwan.

**Writing – review & editing:** Shikha Rimal, Krishna Chandra Ojha, Warangkhana Chaisowwong, Yogendra Shah, Dhan Kumar Pant, Anucha Sirimalaisuwan.

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
