## [Decision Letter · Decision Letter 0]

27 Jan 2020

PONE-D-19-32427

Factors Associated with Rabies Antibody Titer Levels 1 of Vaccinated Dogs in Kathmandu Valley, Nepal

PLOS ONE

Dear Dr Shah

Thank you for submitting your manuscript to PLOS ONE. After careful consideration, we feel that it has merit but does not fully meet PLOS ONE’s publication criteria as it currently stands. Therefore, we invite you to submit a revised version of the manuscript that addresses the points raised during the review process.

:

Many thanks for submitting your manuscript to PLOS One

The manuscript was reviewed by two experts in the field, and they have suggested some minor modifications be made prior to acceptance

Reviewers have made a number of suggestions, which a good response to reviewers will help to speed up re-review

Where reviewer two has made minor grammatical revisions, at their suggestion, these can just be done and not included in the responses to reviewers

I wish you the best of luck with your revisions

Many thanks

Simon

We would appreciate receiving your revised manuscript by Mar 12 2020 11:59PM. To enhance the reproducibility of your results, we recommend that if applicable you deposit your laboratory protocols in protocols.io, where a protocol can be assigned its own identifier (DOI) such that it can be cited independently in the future. For instructions see: http://journals.plos.org/plosone/s/submission-guidelines#loc-laboratory-protocols

We look forward to receiving your revised manuscript.

Kind regards,

Simon Russell Clegg, PhD

Academic Editor

PLOS ONE

Journal Requirements:

3. In your Methods section, please provide additional information on the animal research and ensure you have included details on : (1) whether consent was obtained from the owners of the pet dogs (2) methods of anesthesia and/or analgesia for blood sampling or other efforts to alleviate animal suffering.

5. Please include your tables as part of your main manuscript and remove the individual files. Please note that supplementary tables (should remain/ be uploaded) as separate "supporting information" files.

Reviewers' comments:

Reviewer's Responses to Questions

**Comments to the Author**

1. Is the manuscript technically sound, and do the data support the conclusions?

Reviewer #1: Yes

Reviewer #2: Yes

2. Has the statistical analysis been performed appropriately and rigorously? 

Reviewer #1: Yes

Reviewer #2: Yes

3. Have the authors made all data underlying the findings in their manuscript fully available?

Reviewer #1: Yes

Reviewer #2: Yes

4. Is the manuscript presented in an intelligible fashion and written in standard English?

Reviewer #1: Yes

Reviewer #2: Yes

5. Review Comments to the Author

Reviewer #1: Overall, the manuscript is good and worth publishing. However, it needs refining and edition to make the information clear. One of the important point to change would be to align the aim of the study and the title of the study.

Reviewer #2: This is a very well written, and interesting article, which I thoroughly enjoyed reading. I have made a few minor comments which are mainly grammatical, but also have a few questions regarding the study and interpretation. Most of the comments are minor. I anticipate being asked to review this again, so if it is a minor grammatical change, please don’t feel that you need to write a big long rebuttal.

Line 21- I am not a fan of using first person words like our in scientific writing. Maybe change this to this study identified….. ?

Line 27- you can remove the word out, as separated is sufficient

Lines 35-37- you talk about seroconversion levels. I do wonder if antibody titre would be better as undetectable seroconversion levels suggests that the animal hasn’t seroconverted?

Line 42- you talk about the Nepalese vaccine, and it almost sounds as if you expect it not to be as good as international vaccines. Is there a reason for that?

Line 50- a high level of rabies antibody titre sounds strange- please reword

Line 55- comma after preventable, and after neglected will help flow

Line 58- delete out

Line 61- a semi colon after India may aid flow

Line 66- a comma after There may aid flow

Line 74- you talk about spill over from stray and community dogs- is there any spill over from wildlife?

Line 78- you talk about high level vaccination coverage- is this in Nepal or elsewhere? It would be nice to clarify that to make it clearer

Line 98- as part of the questionnaire, you ask about recipient of vaccination. In the Uk and US you get a confirmation of vaccination. Is that available in Nepal? And if not, how do you know that they were telling the truth?

Line 103- How was the blood separated out?

Line 109- you mention the specificity and sensitivity of the assay- is this done as part of the study or is it what the manufacturer has recommended? Please state that in the manuscript

Line 110- sample dilution- how was this done? In what? And to what concentration?

Line 112- 3 needs to be in full, and it would be three washing steps (rather than washings)

Line 114- 5 needs to be in full, and it would read better as five washing steps

Line 115- comma after well

Line17- what is 1N- should this be 1M for molarity?

Line 149- required may be better than requested

Line 149-151- again seroconversion may be better written as antibody titres?

Line 154- It would be very good to name the vaccines in here as the company, and the people would benefit from knowing from this relative lack of efficacy.

You also talk about three different types of vaccine- are they different makes, killed, modified live? A bit more detail in here would be useful

Line 155-156- this reads a bit unclearly, but I cannot work out a better way to put it

Line 159- you discuss vaccine failures, but I think to call it a failure is difficult due to so many potential reasons why they may not work. Poor administration, out of date, poor storage etc. Maybe tone this down, or discuss reasons why they may have failed

Line 161- 7 in words

Line 162-163- it is reported that black and tan dogs respond poorly, particularly to parvovirus vaccines. Is there any evidence of this here?

Line 170- you analyse 28 different factors but don’t mention which ones. It would be good to have some idea of what you analysed

Line 174- again, We and our would be better replaced with non first person words

Line 179- can remove the P. from P. Singh

Line 179 … showed only 16% of household ….

Lines 177-183- you discuss different studies which have different antibody titres. How many of these are vaccine induced? Or doesn’t it say?

Line 186- 3 in words

Line 189- as stated above, I feel calling it a vaccine failure is a bit misleading, and maybe need quantifying.

Line 193-196- a bit more detail on the vaccines and how they differ maybe useful here- strains, administration, DOI, boosters, preparation etc

Line 197- 2 should be in words

Line 197- How much of the vaccine failures could be associated with MDA? Could this be a cause of poor vaccine response in dogs under 2 years?

Line 202- again, replace we

Line 208- But in this study, no relationship ….

Line 212- again, replace we

Line 213- I have not encountered a coronavirus vaccine before? Is this common in Nepal? Is parvovirus not commonly vaccinated against?

Line 216 this reads a bit unclearly, consider rewording.

Line 218- 2 needs to be in words

Line 219- titre testing. Is there commonly done post vaccination in Nepal? Maybe this could be a suggestion for the conclusions?

Line 220- again replace we

Line 222- outdoor dogs would also have chance to interact with wildlife. Is this considered a risk for infection for these animals?

Line 227-230- you discuss a low IgG response in some animals. How many animals is this issue thought to affect? Could it be a cause of your poor vaccine response?

Line 237- decreased may sound better than less?

Line 239- exposure instead of exposer

Line 248- again, replace we

Line 249- this seems a strange comment considering that you have said that vaccine B gives a poor response?

Line 251- does anti rabies vaccination need capitals?

Line 252- the alteration from bi-yearly to yearly vaccination is an interesting idea. How does this work with costs? And regulatory bodies?

Line 253- how can yearly vaccination be as economically viable as bi-yearly? You need 2 vaccines instead of one which will cost more?

Line 257- again, replace we

Line 258- comma after simultaneously

Line 260 again replace we

Line 261- you say here that you collected samples from clinics, but the methods implies more that you go out to houses and test. Could you make this clearer in the methods?

Could you remove some of the figure legends, as they are in 3 or 4 times?

Figure 1-= a map of where these areas are in Nepal may be useful as my Nepalese geography is terrible. Is there any differences between the areas?

Figure 2- it would be good to label which bit means vaccines, and what the titre is, because these values are low and you mention over 120 in the text which doesn’t link to the figure

Table 1- your second line adds up to 89%- where is the other 11% . Seroconverted could be one word too

Table 2- a bit more detail in the table legend as to what all the abbreviations means may be helpful too

6. PLOS authors have the option to publish the peer review history of their article (what does this mean?). If published, this will include your full peer review and any attached files.

Reviewer #1: No

Reviewer #2: No

---

## [Author Response · Author response to Decision Letter 0]

17 Mar 2020

Comments to authors

Lines 17 and 19: In line 17, you have mentioned rabies impacts heavily on rural but in line 19, there is mention of dogs being the principal vector for rabies transmission in the urban areas. There is a disconnect in the flow of information. Could you please make it clearer. Also, how rabies impacts heavily in the rural and what is the principal vector for rabies in the rural area? Same goes for lines 45 and 46 under author summary.

Response: We appreciate your comments. To make the sentences more coherence and follow information, we have modified the sentences in lines 17, 19, 45 and 46 which are highlighted by red color for your kind information.

Line 21: Here, you have mentioned that the study aimed to measure the efficacy of vaccination …….. but the title of this article is “Factors Associated with Rabies Antibody Titer Levels of Vaccinated Dogs…….”. to me, the aim of the study and title do not match. In the aim, it seems that to evaluate vaccine efficacy was your primary investigation while the factors to be secondary. However, study title indicates factors for vaccine efficacy to be your primary investigation. 

Response: Thank you so much for your comments. We have changed title and aim of study as per your suggestions.

Lines 55 and 56: Please include references.

Response: As per your suggestions, we have incorporated reference in lines 55 and 56 that is highlighted by red color.

Line 68 and 69: Please clarify and elaborate as to what is meant by “true status”.

Response: Thank you for your comment “ true status” meant rabies burden in Nepal which has been corrected in the lines 68 and 69.

Lines 74, 75 and 76: High rabies cases in humans in Nepal is due to more stray and community dogs or due to more rabies in the pets? Please clarify. 

What is the overall prevalence of rabies or prevalence in pet, stray and community dogs? If you have this data then the burden of human rabies due to dogs can be explained. 

Response: Thank you for your comments. Host wise prevalence study also suggests that 61.7% rabies cases were detected in stray and community dogs (Pant et al, 2013). Therefore, high human rabies cases are due to stray and community dogs in Nepal. We have added one supporting reference in line 74, 75 and 76.

Lines 78, 79 and 80: could you please rephrase the sentence to make the point clear. 

Response: As per your suggestions, we rephrased in lines 78, 79 and 80 for your kind information that is indicated by red color.

Lines 90 and 91: Please clarify your aim of the study and the title of the study. Your title says factors associated …… but your says different. 

Response: Thank you very much for your kind comments. We have corrected it as per your previous suggestions mentioned in line 21.

Lines 92 and 93: Are the pet dogs vaccinated as a campaign program or as and when requested by the owner? 

Response: Thank you very much for your comments. In Nepal, the pet dogs are vaccinated both in the campaign program as well as requested of dog owners.

Line 218 and 219: Here, it is mentioned that the insufficient titres could be due to recent vaccination. But I understand from the inclusion criteria that the blood collection was done one month after the last booster. It would be clear if you had data on the number of dogs that were vaccinated recently. In addition, you could also look for information on the non-respondents to vaccination as some animals innately do not respond irrespective of time since vaccination and type of vaccines. 

Response: As per your suggestions, we have incorporated influencing factors for insufficient production of antibody titer level. The factor like age, breeds, type of vaccines could be influencing factor. We have also inserted supporting information and reference.

Lines 228-230: Please add references.

Response: As per your suggestions, we have inserted reference number 23 in line 228-230 for your kind information.

Lines 231-244: The paragraph is informative but does not quite connect with the present study.

Response: We appreciate your comments and we also realized that the paragraph is more informative but irrelevant in this study so, we removed it from this manuscript.

Reviewer #1: Overall, the manuscript is good and worth publishing. However, it needs refining and edition to make the information clear. One of the important point to change would be to align the aim of the study and the title of the study.

Response: Thank you very much for your comments. We have corrected all comments assigned by reviewer 1.

Reviewer #2: This is a very well written, and interesting article, which I thoroughly enjoyed reading. I have made a few minor comments which are mainly grammatical, but also have a few questions regarding the study and interpretation. Most of the comments are minor. I anticipate being asked to review this again, so if it is a minor grammatical change, please don’t feel that you need to write a big long rebuttal.

Line 21- I am not a fan of using first person words like our in scientific writing. Maybe change this to this study identified….. ?

Response: Thank you very much for your kind comments. We have changed first personal pronoun “our” into this study for your kind information.

Line 27- you can remove the word out, as separated is sufficient

Response: As per your suggestions, we removed “out” from the line 27 that is highlighted by red color.

Lines 35-37- you talk about seroconversion levels. I do wonder if antibody titre would be better as undetectable seroconversion levels suggests that the animal hasn’t seroconverted?

Response: Thank you very much for your kind comments. We followed WHO guidelines. According to WHO, the recommended level of protection is (≥ 0.5 IU/ml) but we used (PlateliaTM Rabies II kit ad usum Veterinarium, Biorad, China. This test had 98.6% specificity and 88.8% sensitivity. For detecting antibody titer level there are other gold standard methods like rapid fluorescent focus inhibition test (RFFIT) it can detect seroconversion level of rabies in dogs.

Line 42- you talk about the Nepalese vaccine, and it almost sounds as if you expect it not to be as good as international vaccines. Is there a reason for that?

Response: In Nepal, cell culture rabies vaccine was produced by rabies vaccine production laboratory (RVPL) and animal trials were done in 2002. Since then, vaccine efficacy has not been conducted till date. This is our first comparative efficacy trials of rabies vaccination in Nepal. This study will generate base line data for the control and eliminated rabies in Nepal.

Line 50- a high level of rabies antibody titre sounds strange- please reword

Response: We have changed high level of rabies antibody titer in line 50 that is highlighted by red color.

Line 55- comma after preventable, and after neglected will help flow

Response: We did it.

Line 58- delete out

Response: Thank you sir, we deleted out from manuscript.

Line 61- a semi colon after India may aid flow

Response: We have put a semi colon after India in text that is highlighted by red color.

Line 66- a comma after There may aid flow

Response: We did it for your kind information.

Line 74- you talk about spill over from stray and community dogs- is there any spill over from wildlife?

Response: Thank you for your comments. Sylvatic rabies cycle exists in Nepal but human rabies cases spilled from wild life animal has not been reported yet.

 Line 78- you talk about high level vaccination coverage- is this in Nepal or elsewhere? It would be nice to clarify that to make it clearer

Response: Thank you very much for your comments. It is WHO recommendation and we have modified these sentences as per reviewers 1 comments and inserted reference also for your kind information.

Line 98- as part of the questionnaire, you ask about recipient of vaccination. In the Uk and US you get a confirmation of vaccination. Is that available in Nepal? And if not, how do you know that they were telling the truth?

Response: Thank for your comments. Dog owners are provided receipt of vaccination cards of their dogs in Nepal. Before blood sample collection, we had checked vaccination cards in details.

Line 103- How was the blood separated out?

Response: We have inserted blood separation method in methodology parts.

Line 109- you mention the specificity and sensitivity of the assay- is this done as part of the study or is it what the manufacturer has recommended? Please state that in the manuscript

Response: The manufacturers company has mentioned specificity and sensitivity in the kits 98.6% specificity and 88.8% sensitivity. We have also mentioned its sensitivity and specificity in manuscript as per you suggestions.

Line 110- sample dilution- how was this done? In what? And to what concentration?

Response: Thank you very much. The sample dilution was done 1/10 ratio (10 μl of sample in 990 μl of dilution solution).

Line 112- 3 needs to be in full, and it would be three washing steps (rather than washings)

Response: We have corrected it.

Line 114- 5 needs to be in full, and it would read better as five washing steps

Response: We have corrected it.

Line 115- comma after well

Response: We have corrected it.

Line17- what is 1N- should this be 1M for molarity?

Response: 1 N mean Normal solution of sulphuric acid (H2SO4) which was provide in test kit.

Line 149- required may be better than requested

Response: We have changed it.

Line 149-151- again seroconversion may be better written as antibody titres?

Response: We have corrected it.

Line 154- It would be very good to name the vaccines in here as the company, and the people would benefit from knowing from this relative lack of efficacy.

You also talk about three different types of vaccine- are they different makes, killed, modified live? A bit more detail in here would be useful

Response: We have mentioned vaccine details with company name in manuscript.

Line 155-156- this reads a bit unclearly, but I cannot work out a better way to put it

Response: We have changed it as per your suggestions. 

Line 159- you discuss vaccine failures, but I think to call it a failure is difficult due to so many potential reasons why they may not work. Poor administration, out of date, poor storage etc. Maybe tone this down, or discuss reasons why they may have failed

Response: Thank you very much for your comments. We have discussed the influencing factors associated with vaccine failures in manuscript.

Line 161- 7 in words

Response: We have corrected it.

Line 162-163- it is reported that black and tan dogs respond poorly, particularly to parvovirus vaccines. Is there any evidence of this here?

Response: This is not any evidence of poor vaccine efficacy of parvovirus reported in Nepal.

Line 170- you analyse 28 different factors but don’t mention which ones. It would be good to have some idea of what you analysed

Response: Thank you for the comments. We have analyzed 28 different factors which are described in the methodology parts. Only four influencing factors were associated with antibody titer and these are mentioned in the analysis table.

Line 174- again, We and our would be better replaced with non first person words

Response: We have corrected it.

Line 179- can remove the P. from P. Singh

Response: We have corrected it.

Line 179 … showed only 16% of household ….

Response: We have corrected it.

Lines 177-183- you discuss different studies which have different antibody titres. How many of these are vaccine induced? Or doesn’t it say?

Response: Thank you for your comments. Details vaccination status and administrated doses has not clearly mentioned in that reference paper.

Line 186- 3 in words

 Response: We have corrected it.

Line 189- as stated above, I feel calling it a vaccine failure is a bit misleading, and maybe need quantifying.

Response: Thank you. We have corrected it Sir. 

Line 193-196- a bit more detail on the vaccines and how they differ maybe useful here- strains, administration, DOI, boosters, preparation etc

Response: Three of the most popular vaccines such as Defensor (Killed Virus), Biocan R (Inactivated Strain) and NeJa Rab (Inactivated Cell culture rabies Vaccine) were included in the research. All of the vaccines are administered subcutaneously that have 1 ml capacity vial. Yearly booster of the vaccine is given. 

Line 197- 2 should be in words

Response: We have corrected it.

Line 197- How much of the vaccine failures could be associated with MDA? Could this be a cause of poor vaccine response in dogs under 2 years?

Response: Only one dog included in the study was four months of age, the vaccine failure of which might be associated with MDA.

Line 202- again, replace we

Response: We have corrected it.

Line 208- But in this study, no relationship ….

Response: We have corrected it.

Line 212- again, replace we

Response: We have corrected it.

Line 213- I have not encountered a coronavirus vaccine before? Is this common in Nepal? Is parvovirus not commonly vaccinated against?

Response: The core vaccination of pets in Kathmandu valley comprises of Anti Rabies Vaccine and Canine Distemper combined vaccine (DHPPiL) which includes Canine Distemper, Canine Hepatitis, Parvo Virus Vaccine, Para Influenza Vaccine and Leptospira Vaccine. Parvo virus vaccination is included in the core vaccination. Corona Virus vaccination is an optional vaccination. 

Line 216 this reads a bit unclearly, consider rewording.

Response: We have corrected it.

Line 218- 2 needs to be in words

Response: We have corrected it.

Line 219- titre testing. Is there commonly done post vaccination in Nepal? Maybe this could be a suggestion for the conclusions?

Response: Titer testing is not commonly done after post vaccination in Nepal. We have recommended it in our conclusion part. 

Line 220- again replace we

Response: We have corrected it.

Line 222- outdoor dogs would also have chance to interact with wildlife. Is this considered a risk for infection for these animals?

Response: Dogs that were not kept inside the home had a greater chance to interact with stray dogs and wild life that could be risk of infection for other animals as well as humans.

Line 227-230- you discuss a low IgG response in some animals. How many animals is this issue thought to affect? Could it be a cause of your poor vaccine response?

Response: Poor vaccine response had reported in twelve dogs in this study it could be cause due to low IgG titer level.

Line 237- decreased may sound better than less?

Response: We have removed this paragraph according to previous reviewer comments.

Line 239- exposure instead of exposer

Response: According to first reviewers suggestion, we removed this paragraph from 231-244.

Line 248- again, replace we

Response: We have corrected it.

Line 249- this seems a strange comment considering that you have said that vaccine B gives a poor response?

Response: We have corrected it.

Line 251- does anti rabies vaccination need capitals?

Response: Generally free rabies vaccination program conducted by government, private clinics, kennel club and other NGOs do not charge for this vaccine in the world rabies day (September 28 of each year). Other than that, private clinic and government hospital charge for anti rabies vaccination in regular vaccination schedule.

Line 252- the alteration from bi-yearly to yearly vaccination is an interesting idea. How does this work with costs? And regulatory bodies?

Response: Usually a yearly booster vaccine is given. At places, local technicians tend to give a booster vaccination every six months. The cost for the pet owners is more as they pay double the money for the same vaccine. We do not have a regulatory body that monitors this malpractice.

Line 253- how can yearly vaccination be as economically viable as bi-yearly? You need 2 vaccines instead of one which will cost more?

Response: Yearly vaccination is more cost effective than bi-yearly. The cost of single dose vaccine is less by half than two doses of the same vaccine. 

Line 257- again, replace we

Response: We have corrected it.

Line 258- comma after simultaneously

Response: We did it. 

Line 260 again replace we

Response: We have corrected it.

Line 261- you say here that you collected samples from clinics, but the methods implies more that you go out to houses and test. Could you make this clearer in the methods?

Response: We corrected this sentences linking with methodology part.

Could you remove some of the figure legends, as they are in 3 or 4 times?

Response: We have corrected it.

Figure 1-= a map of where these areas are in Nepal may be useful as my Nepalese geography is terrible. Is there any differences between the areas?

Response: We have included an area map as per your suggestions. 

Figure 2- it would be good to label which bit means vaccines, and what the titre is, because these values are low and you mention over 120 in the text which doesn’t link to the figure

Table 1- your second line adds up to 89%- where is the other 11% . Seroconverted could be one word too

Response: We have added as per your suggestions.

Table 2- a bit more detail in the table legend as to what all the abbreviations means may be helpful too

Response: We have added as per your suggestions.

---

## [Decision Letter · Decision Letter 1]

25 Mar 2020

PONE-D-19-32427R1

Detection of virus-neutralizing antibodies and associated factors against rabies in the vaccinated household dogs of Kathmandu Valley, Nepal

PLOS ONE

Dear Dr Shah

Thank you for submitting your manuscript to PLOS ONE. A few very minor changes have been recommended, These are mainly just grammatical points, which will expedite publication, and will only take around 10 minutes to complete. Therefore, we invite you to submit a revised version of the manuscript that addresses the points raised during the review process.

Many thanks for resubmitting your manuscript to PLOS One

The expert reviewers are happy. I have reviewed the manuscript and highlighted a few minor grammatical issues

If you could modify these, I can recommend it for publication without need for another review

This will hopefully expedite publication at the editorial office

Please do not write a detailed response to reviewers, just a single line saying that the changes have all been made will suffice.

I wish you good luck with your modifications

Hope you are keeping safe in these difficult times

Thanks

Simon

We would appreciate receiving your revised manuscript by May 09 2020 11:59PM. To enhance the reproducibility of your results, we recommend that if applicable you deposit your laboratory protocols in protocols.io, where a protocol can be assigned its own identifier (DOI) such that it can be cited independently in the future. For instructions see: http://journals.plos.org/plosone/s/submission-guidelines#loc-laboratory-protocols

A marked-up copy of your manuscript that highlights changes made to the original version. This file should be uploaded as separate file and labeled 'Revised Manuscript with Track Changes'.An unmarked version of your revised paper without tracked changes. This file should be uploaded as separate file and labeled 'Manuscript'.

We look forward to receiving your revised manuscript.

Kind regards,

Simon Russell Clegg, PhD

Academic Editor

PLOS ONE

Reviewers' comments:

Reviewer's Responses to Questions

**Comments to the Author**

1. If the authors have adequately addressed your comments raised in a previous round of review and you feel that this manuscript is now acceptable for publication, you may indicate that here to bypass the “Comments to the Author” section, enter your conflict of interest statement in the “Confidential to Editor” section, and submit your "Accept" recommendation.

Reviewer #1: All comments have been addressed

Reviewer #2: All comments have been addressed

2. Is the manuscript technically sound, and do the data support the conclusions?

Reviewer #1: Yes

Reviewer #2: Yes

3. Has the statistical analysis been performed appropriately and rigorously? 

Reviewer #1: Yes

Reviewer #2: Yes

4. Have the authors made all data underlying the findings in their manuscript fully available?

Reviewer #1: Yes

Reviewer #2: Yes

5. Is the manuscript presented in an intelligible fashion and written in standard English?

Reviewer #1: Yes

Reviewer #2: Yes

6. Review Comments to the Author

Reviewer #1: (No Response)

Reviewer #2: I have reviewed your manuscript to ensure that it is ready to go for publication. I have made a few minor suggestions for grammatical changes which will increase the flow of the reading. It may look like there is a lot, but it will probably only take you 10 minutes maximum.

This will hopefully increase the speed of publication when it gets to the editorial office.

Line 22- …the aim of detecting virus-neutralising…..

Line 36- …(<125 EU/ml) suggesting that the ….

Line 42- …. manufactured brand of vaccine, which is available in Nepal, is potent …

Line 49- …efficacy of vaccination programmes, and to determine whether ….

Line 57- ….is a vaccine preventable, viral zoonosis ….

Line 58- …all human deaths from rabies are caused ….

Line 59- …59,000 people due from this disease …

Line 60- …of which 30-50% are below ….

Line 61- …60% of the total number of cases ….

Line 70- … the year, and dogs are ….

Line 73 -…involves domesticated dogs, and the other is ….

Line 76- Large numbers of human rabies cases …..

Line 82- For the dog rabies mass vaccination program, achieving a ….

Line 89-… vaccines been used, or to determine if the ….

Line 98- …were taken from the Nepal Veterinary ….

Line 102- agriculture doesn’t need to be capitalised

Lines 98-104- this doesn’t need to be in bold

Line 107- ….districts of Kathmandu valley, shown in map….

Line 108- Inclusion criteria were a dog ages above 4 months ….

Line 120- Biorad, China) was used, following the ….

Line 131- adding to each well, a peroxidase substrate …..

Line 135- ….quantification standards (S1 to S6) , obtained by serial ….

Line 137- …titers in quantification tests, obtained after a ….

Line 143- ….regarding each pet dog (age, sex ….

Line 162- …optical density compared to a standard curve ….

Line 173- please put 3 in words

Line 175. However, there was no significant ….

Line 178- …less than two years old (47%) …

Line 193- …antibody was detected in 110 pet dogs….

Line 194- …89.09% had the necessary ….

Line 195- …..(>0.5 IU/ml), which is in line ….

Line 207- …may be due to the age of the vaccinated ….

Line 217- Higher numbers of vaccine failures ….

Line 218- …in younger dogs, because their ….

Line 219- ….has been documented [19], which may ….

Line 222- …analysis, however, no significant relationship….

Line 223- dog and their titer level was found

Line 228- double space between higher and failure (please remove 1)

Line 232- double space between give and core (please remove 1)

Line 232- …which includes the anti-rabies. …

Line 233- Canine distemper doesn’t need capitals

Line 234- diseases from canine distemper to leptospirosis don’t need capitals

It may also be worth mentioning that there is, as yet, no studies on variation in canine parvovirus published from Nepal. This is one of the diseases which I work on so it maybe worth a chat in the future to get some samples and do this study?

Line 235- coronavirus doesn’t need a capital

Line 235- However, there was no….

Line 236 and 237- Anti-rabies, corona, canine and distemper, don’t need capitals

Line 242- ….insufficient titer level, while two dogs ….

Line 251- wildlife is one word

Line 253- Due to some health conditions, prior to the vaccination….

Line 257- …does not produce sufficient, specific IgG….

Line 262- However, in the market, there is a tendency ….

Line 266- …safely said that anti rabies vaccination ….

Line 273- Titre testing post vaccination …..

7. PLOS authors have the option to publish the peer review history of their article (what does this mean?). If published, this will include your full peer review and any attached files.

Reviewer #1: No

Reviewer #2: Yes: Simon Clegg

---

## [Author Response · Author response to Decision Letter 1]

2 Apr 2020

Comments to authors

Reviewer #1: (No Response)

Reviewer #2: I have reviewed your manuscript to ensure that it is ready to go for publication. I have made a few minor suggestions for grammatical changes which will increase the flow of the reading. It may look like there is a lot, but it will probably only take you 10 minutes maximum.

This will hopefully increase the speed of publication when it gets to the editorial office.

Response: Thank you very much sir for your kind valuable suggestions and comments for grammatical changes to enhance our revised manuscript flow of the reading. As per your kind suggestions, we have modified and made all single inline grammatical changes in revised manuscript with red color according to assigned by reviewer 2 for your kind information

---

## [Editor Report · Decision Letter 2]

6 Apr 2020

Detection of virus-neutralising antibodies and associated factors against rabies in the vaccinated household dogs of Kathmandu Valley, Nepal

PONE-D-19-32427R2

Dear Dr. Shah

We are pleased to inform you that your manuscript has been judged scientifically suitable for publication and will be formally accepted for publication once it complies with all outstanding technical requirements.

With kind regards,

Simon Russell Clegg, PhD

Academic Editor

PLOS ONE

Additional Editor Comments (optional):

Many thanks for submitting your manuscript to PLOS One

I have reviewed your manuscript. Many thanks for making the suggested changes.

I have recommended your manuscript for publication, so you should hear from the editorial office soon.

There are two minor changes which need to be made, and if these could be made during editing for publication, I would be very grateful

Line 59- change due to die

Line 88- change been to being

I wish you all the best for your future research. I also recommend looking into a study on canine parvovirus in Nepal, and if I can be of any assistance with this then please let me know as its one of my research interests

Please stay safe and well during the current difficult times

Thanks

Simon

---

## [Editor Report · Acceptance letter]

10 Apr 2020

PONE-D-19-32427R2 

Detection of virus-neutralising antibodies and associated factors against rabies in the vaccinated household dogs of Kathmandu Valley, Nepal 

Dear Dr. Shah:

I am pleased to inform you that your manuscript has been deemed suitable for publication in PLOS ONE. Congratulations! Your manuscript is now with our production department. 

With kind regards,

on behalf of

Dr. Simon Russell Clegg 

Academic Editor

PLOS ONE